# The Use of Electrochemical Impedance Spectroscopy as a Tool for the In-Situ Monitoring and Characterization of Carbon Nanotube Aqueous Dispersions

**DOI:** 10.3390/nano12244427

**Published:** 2022-12-12

**Authors:** Aikaterini Gkaravela, Ioanna Vareli, Dimitrios G. Bekas, Nektaria-Marianthi Barkoula, Alkiviadis S. Paipetis

**Affiliations:** Department of Materials Science and Engineering, University of Ioannina, 45110 Ioannina, Greece

**Keywords:** Electrochemical Impedance Spectroscopy (EIS), carbon nanotubes (CNTs), dispersion, sonication process

## Abstract

So far, there is no validated technology for characterizing the dispersion and morphology state of carbon nanotubes (CNTs) aqueous dispersions during sonication. Taking advantage of the conductive nature of CNTs, the main hypothesis of the current study is that Electrochemical Impedance Spectroscopy (EIS) is an appropriate technique for the in-situ monitoring and qualification of the dispersion state of CNTs in aqueous media. To confirm our hypothesis, we monitored the Impedance |Z| during the sonication process as a function of type CNTs/admixtures used for the preparation of the aqueous solutions and of crucial process parameters, such as the applied sonication power and duration (i.e., sonication energy). For dispersions above the percolation threshold, a drop of |Z| by approximately seven orders of magnitude was observed, followed by a linear reduction. The dramatic change in |Z| is regarded as an indication of the formation of a conductive path or destruction of an existing one during sonication and can be used to characterize the dispersion and morphology state of CNTs. The results of the EIS provide, straightforwardly and reliably, the required information to create an optimum dispersion protocol for conductive CNT suspensions. The produced dispersions are part of research focusing on the manufacturing of cement-based composite materials with advanced thermoelectric functionalities for energy harvesting. Such dispersions are not only limited to energy harvesting applications but also to applications where functionalities are introduced through the use of conductive-based suspensions.

## 1. Introduction

One approach to achieving next-generation smart materials is incorporating nanoinclusions, such as carbon nanotubes (CNTs), into matrices to produce composite materials for multifunctional purposes [1,2,3,4]. Due to their remarkable intrinsic properties (high stiffness, strength, thermal and electrical conductivity, low density, etc.) CNTs are materials that could find application in many areas of technology [5]. It is widely accepted that adding well-dispersed CNTs into a matrix could not only enhance its mechanical performance (e.g., toughness), but could also increase its electrical and thermal conductivity and, in turn, its ability to interact with external stimuli, harvest energy, etc. [6,7,8,9]. An optimized and efficient dispersion significantly contributes to cost reduction. If the dispersion of a nanophase within a matrix is enhanced, the amount of the reinforcing phase required to increase the mechanical performance or to achieve the electrical percolation threshold is significantly reduced. This is directly linked to the costs involved with CNT utilization. Additionally, the more efficient a CNT dispersion, the more sustainable the final material would be since the amount of the nanophase required to achieve a certain performance enhancement is significantly decreased [10].

The dispersion of CNTs can be accomplished through their direct addition into the matrix or by adding them to an appropriate solvent, such as water. Aqueous dispersions have recently attracted the interest of the research community since water is very frequently used as a solvent. Furthermore, it is a non-toxic solvent, resulting in composites that are more environmentally compliant [11]. However, the dispersion of the CNTs is challenging since, due to their structure and hydrophobic behavior, CNTs tend to agglomerate and form bundles [12,13]. The use of surfactants, through non-covalent adsorption on CNTs surfaces, facilitates the compensation of the Van der Waals attractions [14] and helps them preserve their dispersion over time, especially in aqueous solutions [15]. To date, various combinations of CNTs/surfactants in aqueous solutions have been investigated [15,16,17]. The amount by weight of the surfactant normally considerably exceeds the amount of the CNTs ranging from 1.5:1 to 10:1, weight ratios, respectively [18,19]. Nevertheless, adding a significant amount of surfactant could lead to deterioration of the properties (e.g., mechanical or electrical) of the final composite [20,21,22]. Therefore, attempts have been made to combine surfactants with a technique that could reduce the self-attraction of CNTs. Ultrasonication is the most adopted technique for the dispersion of CNTs in solutions since it provides high local shear that could de-bundle the CNTs [23,24,25,26]. The separating forces increase the concentration of the dispersed CNTs by detaching individual CNTs from bundles. On the other hand, the scissoring forces reduce the length of the CNTs and, in turn, their aspect ratios. The level of separated and well-dispersed CNTs and their length distribution are the two important factors that must be balanced during ultrasonication to achieve enhanced performance at a minimum CNT concentration [27,28,29,30]. For instance, it has been shown that through the use of ultrasonication it was possible to reduce the surfactant/CNTs ratio to as low as 0.5:1 without considerably affecting the dispersion quality, while the increase of this ratio up to 2:1 reduced the required ultrasonication time for adequate dispersion [31]. However, there is no standard guideline to perform the ultrasonication process since the efficiency of the process depends on many factors, such as the duration of the process, the applied power, the sonicator type (probe, type of probe, bath), the temperature, and of course the type and the amount of the CNTs and/or dispersive agents [31,32].

One major problem in achieving the right balance between sufficient dispersion and maintenance of the length of the CNTs stems from the fact that there is no validated technology for the in-situ monitoring and qualification of the dispersion state in aqueous media. Scanning electron microscopy (SEM) [33], transmission electron microscopy (TEM) [34], optical microscopy [25], light scattering methodologies [31,35], and atomic force microscopy (AFM) [36] have been extensively used to characterize the CNTs dispersion over the past years. Although these methods can estimate the diameter and the length of the CNTs, they require a considerable amount of time for the sample preparation and are mainly applied to a representative volume of the sample as offline monitoring techniques. On the other hand, the use of spectroscopic methods, such as Raman [37] and UV-Vis [38], enables the quantification of the dispersion properties of the CNTs [39]. However, these techniques also require the preparation of the samples and post-processing of the results to evaluate the dispersion state. Furthermore, Raman spectroscopy can be performed only on solid-state samples, light scattering methodologies, and UV-Vis spectroscopy require sample dilution; therefore, some processes are limited to offline monitoring [31,35,37,38].

Electrochemical Impedance Spectroscopy (EIS) has gained considerable attention over the years, especially in polymer- and cement-based composite materials, as it has proved to be a useful tool for the characterization of the microstructure [40,41], detection of damage [42,43,44], monitoring of curing in the polymer matrix [45], hydration in the cement matrix [46], pore structure [47] and for the evaluation of the crack propagation [48,49]. So far, limited information is available in the open literature around the use of EIS for the characterization of the dispersion state and/or confirmation of the percolation of nanomaterials in aqueous solutions. Tsirka et al. [50] performed an in-situ monitoring of the dispersion of oxidized CNTs in an aqueous solution via EIS. They confirmed the existence of electrical percolation after conducting experiments in multiple amounts of oxidized CNTs at different sonication times. On the other hand, the characterization of the dispersion of nanomaterials in epoxy resin has been thoroughly investigated. For instance, Bekas et al. [51], Baltzis et al. [52], and Foteinidis et al. [53] applied EIS. They concluded that by monitoring the dependence between the Impedance |Z| and the dispersion duration, it was possible to achieve valuable information on the dispersion state of CNTs and CNTs/carbon black in epoxy systems.

In the present research, EIS is applied to investigate the dispersion of aqueous solutions with a low surfactant/CNTs ratio, which is highly relevant for preparing functional cementitious composites for energy harvesting without side effects in their physical and electrical properties [54,55]. Since it has been recognized from previous studies [28,29,38] that the ultrasonication energy (combination of time and power) is crucial for the formation of well-distributed CNTs and a continuous, functional network, the current study focuses on the monitoring of |Z| using different sonication parameters. Thus, the influence of key material (i.e., type of CNTs, presence of admixtures) and process parameters (applied ultrasonication power, duration of the ultrasonication, and the sonication probe) on the dispersion quality and monitoring ability of EIS is thoroughly assessed. The obtained results demonstrate for the first time that EIS is an appropriate technique for in situ measurements of the entire dispersion volume capable of rapidly detecting any change at any stage of the dispersion process.

## 2. Materials and Methods

### 2.1. Materials

In this study, two different types of CNTs are used. Multi-wall CNTs (MWCNTs) in powder form under the brand name of Nanocyl were provided by (Nanocyl). The CNTs had an average diameter of 9.5 nm, an average length of 1.5 µm, carbon purity of 90%, and a transition metal oxide < 1%. Single-wall CNTs (SWCNTs), also in powder form, were provided by (OCSiAl). The product name is TUBALL, with the specific characteristics of an average diameter of 1.8 ± 0.4 nm, length > 5 µm, carbon content > 85 wt.%, and metal impurities < 15 wt.%. Sodium dodecylbenzene sulfonate (SDBS), an anionic surfactant, was purchased from (Sigma Aldrich, St. Louis, MO, USA). Polyethyleneimine (PEI) solution ~50% in H_2_O (Mr 600,000–1,000,000) was also purchased from (Sigma Aldrich). All chemical reagents are used as received.

### 2.2. Preparation of CNT-Based Aqueous Dispersions

Aqueous dispersions are prepared using different amounts of CNTs (0.2, 1, and 2 wt.% by water content), while the CNTs/SDBS ratio is kept constant at 1:1. The produced dispersions are part of research focused on the manufacturing of cement-based composite materials with advanced thermoelectric functionalities. Considering a water/cement ratio of 0.5, typical for cementitious materials, the selected CNT contents correspond to 0.1, 0.5, and 1 wt.% of cement. Since the prepared dispersions are relevant not only for cementitious materials but also for the preparation of other solution-based castings, the wt.% of CNTs in the present study will refer to the water content of the prepared dispersions. To produce a thermoelectric generator device (TEG) [2], it is necessary to convert the positive (p-type) semiconductor behavior of CNTs to negative (n-type). This can be achieved by adding a polymeric dopant, e.g., Polyethyleneimine (PEI). When applicable, a PEI-based solution is added to water at a CNT/PEI ratio of 1:5. Previous work from our group indicated that PEI could be effective in converting the behavior of CNTs from p- to n-type CNTs [56], while based on unpublished data, the proposed ratio is the one that resulted in the lowest resistivity values of the converted CNTs.

For the preparation of the CNT-based aqueous dispersions, SDBS is first dissolved in deionized water with the help of a magnetic stirrer. Then CNTs are added, and after approximately 30 min of stirring, the produced solutions are sonicated for durations of 30, 60, and 90 min, respectively, using a UP400S ultrasonic processor supplied by Hielscher Ultrasonics. Two different sonication probes are used, one with a 40 mm diameter, referred to as the big sonication probe, and one with a 7 mm diameter, named the small sonication probe. For the dispersions with the small sonication probe, three different levels of sonication power are applied through the manipulation of the total amplitude capacity of the sonicator (25%~13.4 W, 50%~17.9 W, and 75%~22.7 W). On the other hand, for the dispersions prepared with the big sonication probe, the sonicator is set at 25% amplitude, corresponding to a sonication power of 18.6 W. Sonication is performed in a cooling bath to prevent an increase in the temperature of the mixture during dispersion. In Appendix A, a schematic representation of the dispersion combinations and the sonication process is presented.

### 2.3. Characterization of the Dispersion with Impedance Spectroscopy

For the characterization of the dispersion of the CNTs with EIS, an Advanced Dielectric Thermal Analysis System (DETA-SCOPE) is used, supplied by (ADVISE, Greece). An interdigital sensor, also manufactured by (ADVISE, Greece), is connected to the DETA-SCOPE, and a sinusoidal voltage is applied. The dielectric sensor is placed inside the dispersion solution, and an impedance measurement is performed after a sonication cycle (i.e., 30, 60, and 90 min, respectively). The EIS measurements were performed between the limits of measurement equipment, from 10^−2^ Hz to 10^5^ Hz, with a 10 V amplitude, and started with a 30 s quiet time. Within this frequency range, the accuracy of the measurements is 0.05%. Appendix A provides an overview of the experimental setup used to perform the EIS measurement during sonication. The analysis of the results in this work is based on the Complex Resistance, the Impedance R. From an electrical standpoint, the Impedance, |Z|(ω), of a material at any angular frequency, ω, follows Equation (1) where the resistive impedance Z′(ω) is the real part, and the reactive impedance Z″(ω) is the imaginary part of the impedance, in Ohm.
(1)$$\left|Z\right|\left(\omega \right)=\sqrt{Z^{\prime }{\left(\omega \right)}^{2}+Z^{\prime \prime }{\left(\omega \right)}^{2}}$$

The real and imaginary part Z′(ω) & Z″(ω) of the Impedance Z can be employed to calculate the conductivity in the frequency domain [57]. The aforementioned equation is included in the Appendix A.

### 2.4. Characterization of the Dispersion with Optical Microscopy

The characterization of the dispersions with optical microscopy was performed on representative dispersions containing 1 wt.% SWCNTs to verify the results obtained by EIS. After every circle of the sonication treatment, a drop of the suspension was carefully sucked up with a pipette and dipped into a glass to settle. The characterization of the suspension’s morphology was studied by a Microphot research optical microscope (Nikon Inc. Instrument Group, Tokyo, Japan).

## 3. Results and Discussion

This study aims to achieve a direct connection between the impedance |Z| and the experimental parameters and confirm that through the EIS results it is possible to define the protocol needed for optimizing the dispersion of the CNTs to achieve conductive aqueous solutions. Different amplitudes have been used in the current investigation (Appendix A); however, the following analysis concentrates on dispersions obtained after applying intermediate sonication power. Note that for the big sonication probe the amplitude of 25% results in 18.6 W. This is slightly higher than the sonication power obtained using the small sonication probe at 50% amplitude (17.9 W). Thus, keeping the sonication power almost the same, the effect of the power density will be assessed in the following paragraphs. Results obtained using the small sonication probe at lower (25%), and higher amplitude (75%) are included in the supporting material section.

Although the study was conducted over a range of CNTs, below and above the percolation threshold, the following paragraphs focus on the results obtained after adding 1 wt.% CNTs to the aqueous solutions, expected to be above the percolation threshold for both MWCNTs [58] and SWCNTs [56]. Results for systems with 0.2 and 2 wt.% CNTs are presented in the supporting material section of the current paper. As illustrated in Appendix A, independent of the sonication protocol, the addition of 0.2 wt.% MWCNTs did not result in low |Z|, suggesting the need for higher MWCNT contents to create conductive suspensions. On the other hand, as illustrated in Appendix A, at 0.2 wt.% SWCNTs, a drop of |Z| by seven orders of magnitude is obtained under specific sonication conditions, suggesting the creation of a conductive network. Although this content is considered below the percolation threshold in the case of MWCNTs, SWCNTs differ in their electrical conductivity properties and can form, at lower contents, a continuous network if successfully dispersed [56]. As expected, under appropriate sonication conditions, 2 wt.% CNTs lead in conductive suspensions, as illustrated in Appendix A. It should be noted that the provided power in conjunction with the sonication duration is crucial for achieving a continuous electrical network, as confirmed by EIS, since using the small sonication probe at 25% amplitude is not enough to properly disperse 2 wt.% CNTs (Appendix A).

In Figure 1, the EIS plot of 1 wt.% MWCNTs dispersions at an intermediate sonication amplitude (i.e., 50%), using the small sonication probe, is presented. As aforementioned, this content was expected to be above the percolation threshold, and appropriate sonication should result in the creation of a conductive suspension. As observed in Figure 1, the Impedance |Z| presented very high values during the first 30 min of the sonication process, which for the low frequencies of the diagram are considered unreal as they are within the set-up measurement limits. Furthermore, the formation of a conductive network, demonstrated by a drop of the impedance |Z| by eight orders, was achieved after 60 min of ultrasonication. After 90 min, the dispersion remained conductive with a slight increase (one order of magnitude) in the |Z| value, which suggests a slight deterioration in the CNT structure due to extensive sonication.

As illustrated in Figure 2, applying almost the same sonication power through a big sonication probe results in efficient dispersion of 1 wt.% MWCNTs after 60 min of sonication, while further sonication (90 min) causes a degradation of the CNTs structure and a destruction of the formed electrical network resulting in a sharp increase in the |Z| magnitude. Note that for the big sonication probe the amplitude of 25% results in 18.6 W. This is slightly higher than the sonication power obtained using the small sonication probe at 50% amplitude (17.9 W). Compared to the small sonication probe that slightly deteriorated the structure of MWCNTs after 90 min of sonication (Figure 1), the bigger surface area of the sonotrode, in conjunction with the slight increase of the provided power (0.7 W), was detrimental to the MWCNTs structure (Figure 2).

The small sonication probe proved to be quite efficient in dispersing MWCNTs without excessive deterioration of their structure, therefore an attempt to disperse 1 wt.% SWCNTs using the same sonication conditions was performed. As observed in Figure 3, independent of the sonication duration, the dispersion presents very high |Ζ| values that suggest the absence of an electrically conductive network. It is therefore deduced that the small sonication probe is insufficient to disperse the SWCNTs even after 90 min, since the van der Waals attractions are much stronger in the case of SWCNTs compared to the ones observed in MWCNTs [59].

Due to the nature of the SWCNTs and based on the results of the MWCNTs dispersion, the big sonication probe at 25% amplitude was applied for the dispersion of the same concentration of SWCNTs. The dispersion efficiency of the solution with 1 wt.% SWCNTs is assessed in Figure 4. A homogeneous dispersion was achieved after 60 min of the sonication process, while further sonication (90 min) destroyed the formed electrical network.

This detection of the degradation of the CNTs structure by EIS was verified by (Optical Microscopy). In Figure 5, the images of the optical microscope (20×) of the dispersion of 1 wt.% SWCNTs are presented. After 30 min of the sonication process (Figure 5a), the network was not formed as the presence of aggregates was remarkable. After 60 min (Figure 5b), a homogenized distribution was observed, followed by a reappearance of aggregates (re-agglomeration) after 90 min (Figure 5c) of the sonication process. This re-agglomeration can justify the increase in |Ζ| as the conductive network is clearly disrupted.

Comparing the obtained results of the EIS for both sonication probes, it can be concluded that even if almost equal sonication power is applied to the same CNT-based aqueous solutions, the energy density, which is controlled by the size of the sonication probe, and the sonication duration, is a crucial parameter for the dispersion ability and maintenance of the CNT structure, as highlighted by the EIS results. In line with recent findings using UV and light scattering methodologies [29,36], EIS confirmed that the energy density provided by the surface of the sonication probe is a significant parameter in the dispersion process. As observed in Figure 3 and Figure 4, although similar power levels are applied with the small and the big sonication probe, the latter is more effective in dispersing SWCNTs. It is believed that the bigger surface area of this type of sonotrode leads to a wider distribution of the ultrasound waves and more efficient interruption of the cohesive forces between the SWCNTs.

Since it is very common to use polymer-based admixtures for the preparation of functional coatings, it is of paramount importance to assess whether these substances interfere with the sonication process and if this can be effectively captured by the EIS monitoring technique. These questions are addressed in the following paragraphs concerning the example of TEG-based solutions by adding PEI.

Thus, Figure 6 presents the effect of PEI addition in MWCNTs dispersions above the percolation threshold using the big sonication probe. As observed, the addition of PEI contributes positively to the dispersion of 1 wt.% MWCNTs, since obtained |Z| magnitude was quite low (app. 10^5^ Ohm at 0.01 Hz) after 30 min of sonication, while the conductive behavior was maintained for up to 60 min of the process. A sharp increase (approximately seven orders of magnitude) of the |Z| values up to the limits of the measurement set-up was observed after 90 min, indicating a degradation of the CNTs structure. It can be suggested that the macromolecules of PEI are dispersed between the individual CNTs, supporting their separation. The dispersion ability of the MWCNTs during the presence of PEI with 2 wt. % is presented in the Appendix A.

PEI is also used to convert SWCNTs from p-type to n-type, and its effect on the dispersive ability of solutions with 1 wt.% SWCNTs is shown in Figure 7. Again, a conductive network formed after 60 min of sonication. However, compared to the results obtained without the addition of PEI (Figure 4), |Z| values are over one order of magnitude higher here. This slight increase in |Z| could be linked to a viscosity increase in the prepared solutions due to the presence of PEI. Interestingly, additional sonication time (90 min) resulted in an improvement of the |Z|, leading to similar levels as those achieved after 60 min of sonication in the absence of PEI (see Figure 4). Furthermore, the deterioration of the CNT structure after 90 min of sonication of solutions without PEI (Figure 4) is not observed here. Based on the above, it can be deduced that the addition of PEI not only facilitates the dispersion process but also protects the SWCNTs from excessive breakage during sonication.

In Figure 8a direct comparison of the dispersion ability of the different experimental combinations is attempted based on the obtained EIS response. For this purpose, the values of |Z| at a frequency of 0.05 Hz was selected for both sonication probes. Since, at low frequencies, the impedance |Z| is within the region that follows Ohm’s law, it can be considered proportional to the DC resistance of the dispersions. Sonication combinations that result in conductive solutions are illustrated with an ellipse on the graph. For a comprehensive comparison, along with the results that are presented in Figure 8, data from the results of additional sonication amplitudes that are shown in the supporting material have been used (see Appendix A).

Overall, it can be concluded that through the variations of the impedance |Z|, differences in the dispersion quality of the CNTs can be identified. Intermedium sonication energies, resulting from intermediate sonication power and medium to high sonication duration, are required for the dispersion of 1 wt.% MWCNTs, as observed in Figure 8a, while PEI addition, in conjunction with the big sonotrode, reduces the required energy for effective dispersion. On the other hand, medium to high sonication energies, only after using the big sonication probe, are appropriate for dispersing 1 wt.% SWCNTs (Figure 8b). The relevant results for systems with 0.2 and 2 wt.% CNTs are presented in the Appendix A. The observed differences in the EIS response with the sonication parameters highlight the importance of controlling each dispersion at every step of the process with a technique that provides direct and accurate information during sonication.

Based on the above, the key innovation of the current study is that it demonstrates for the first time how EIS can be successfully employed to monitor and decouple all intricate mechanisms involved in the dynamic process of dispersion of CNTs in aqueous media through the monitoring of the Impedance |Z|.

This is quite advantageous for the process control of the dispersion of carbon-based suspensions at the industrial level over state-of-the-art techniques such as SEM [33], TEM [34], DLS [35], AFM [36], Raman [37], and UV-Vis [38] which are used as offline monitoring methodologies on representative volumes of the material.

The achievement of percolation in dispersions with a low surfactant/CNTs ratio, while maintaining the morphology of CNTs, is a very challenging task. Its success could result in high-performance functional dispersions and, in turn high, performance functional materials that could be used for the preparation of sensors [60], energy harvesting materials [2], wearable electronics [61], etc. Since functional suspensions can be prepared using different fillers, a potential avenue for future work is to confirm the sensitivity and accuracy of EIS’ monitoring capability with other conductive materials used as additives for multifunctional purposes.

## 4. Conclusions

The current study performed, for the first time, a thorough investigation of the monitoring ability of EIS during the dispersion process of both SWCNTs and MWCNTs in aqueous solutions. The study was performed with a low surfactant/CNTs ratio, as a function of key material and process parameters. As was demonstrated, the effect of any experimental parameters can be detected instantaneously with EIS through changes in the Impedance |Z|.

Analytically:

Not only the required energy but also the appropriate combination of power, duration, and energy density are crucial for the effective dispersion of CNTs;

High ultrasonication power and/or time can destroy the structure of the CNTs from the beginning of the ultrasonication process and destroy the already-formed electrical network;

The sonication power provided by the 25% amplitude of the big sonication probe proved to be quite efficient in dispersing both types of CNTs;

The energy density provided by the surface of the sonication probe is a significant parameter in the dispersion process in that very similar sonication energies result in very different dispersion qualities;

The addition of polymer-based admixtures (e.g., PEI) alters the sonication power and time required for efficient dispersion;

Overall, a sonication process depends highly on the suspension parameters under investigation, and an inline monitoring methodology is essential for establishing a relevant protocol. EIS can provide real-time feedback on the dispersion of CNT-based suspensions. Thus, percolation can be achieved without degradation of the nanophase. EIS in-line monitoring can be easily applied at both research and industrial level for efficient and reliable process control.

## Figures and Tables

**Figure 1 nanomaterials-12-04427-f001:**
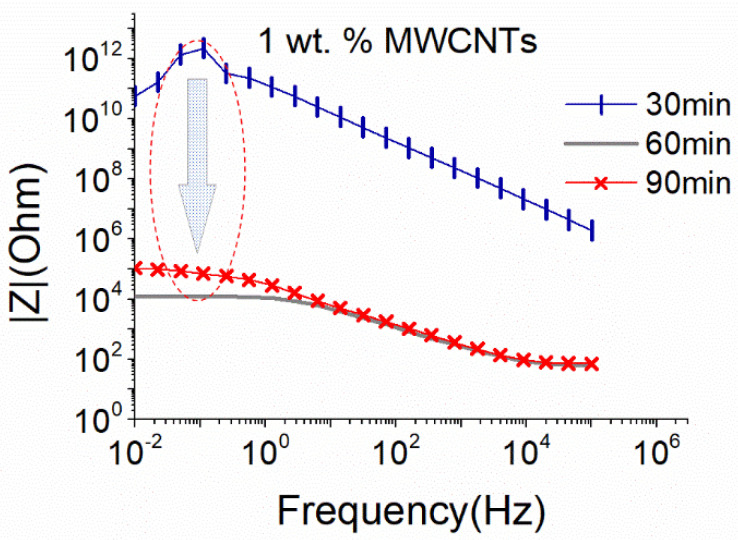
EIS plots of aqueous dispersion with 1 wt.% MWCNTs using the small sonication probe at an amplitude of 50% (17.9 W).

**Figure 2 nanomaterials-12-04427-f002:**
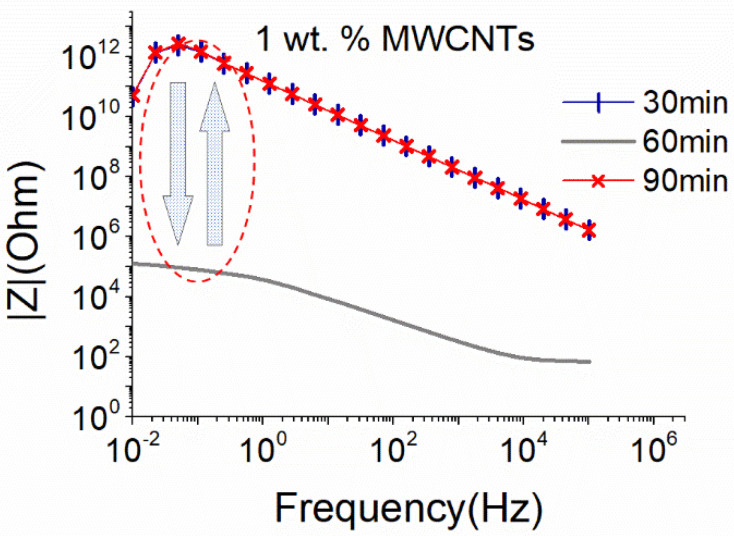
EIS plots of aqueous dispersion with 1 wt.% MWCNTs using the big sonication probe at an amplitude of 25% (18.6 W).

**Figure 3 nanomaterials-12-04427-f003:**
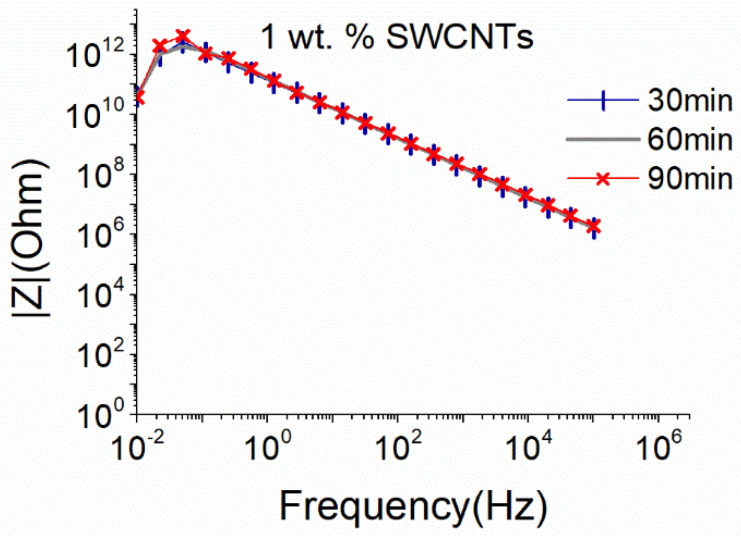
EIS plots of aqueous dispersion with 1 wt.% SWCNTs, using the small sonication probe at an amplitude of 50% (17.9 W).

**Figure 4 nanomaterials-12-04427-f004:**
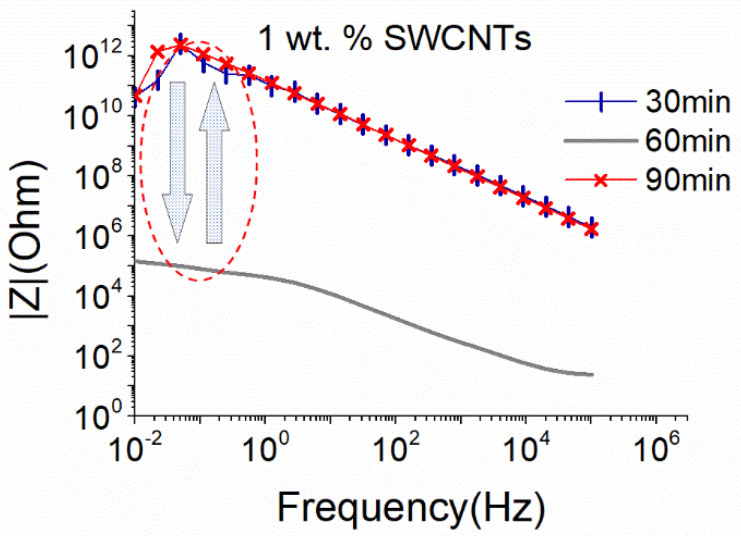
EIS plots of aqueous dispersion with 1 wt.% SWCNTs, using the big sonication probe at an amplitude of 25% (18.6 W).

**Figure 5 nanomaterials-12-04427-f005:**
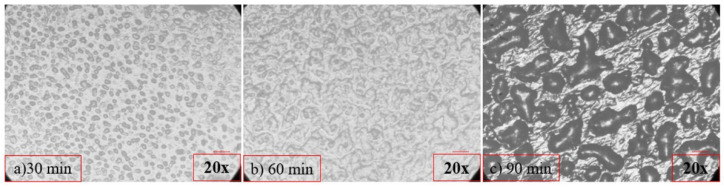
Optical Microscopy images of the suspension containing 1 wt.% SWCNTs after (**a**) 30 min, (**b**) 60 min, and (**c**) 90 min of the sonication process.

**Figure 6 nanomaterials-12-04427-f006:**
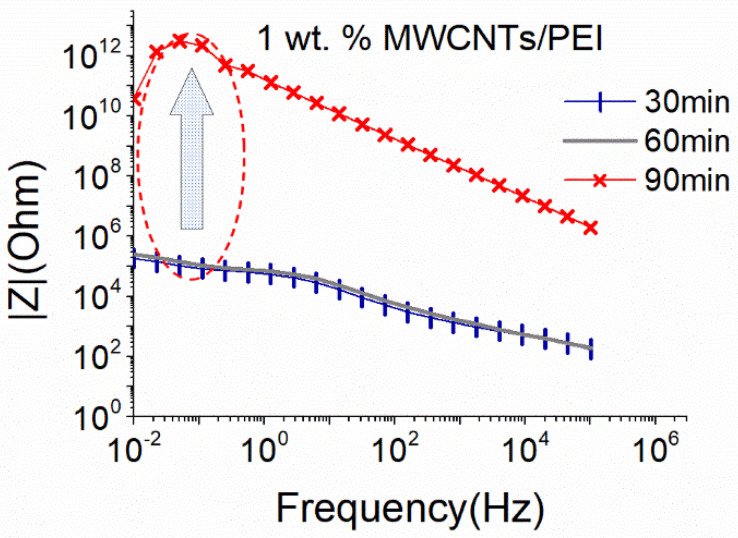
EIS plots of aqueous dispersion with 1% MWCNTs, using the big sonication probe at an amplitude of 25%, in the presence of PEI (18.6 W).

**Figure 7 nanomaterials-12-04427-f007:**
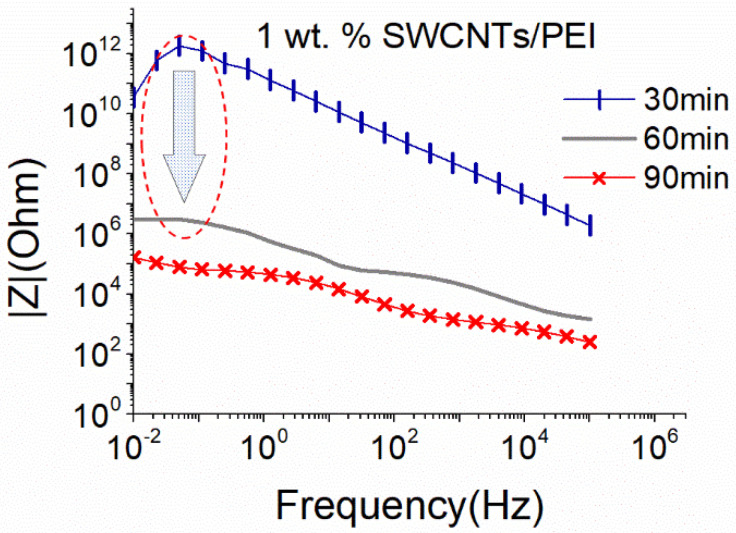
EIS plots of aqueous dispersion with 1 wt.% SWCNTs, using the big sonication probe at an amplitude of 25%, in the presence of PEI (18.6 W).

**Figure 8 nanomaterials-12-04427-f008:**
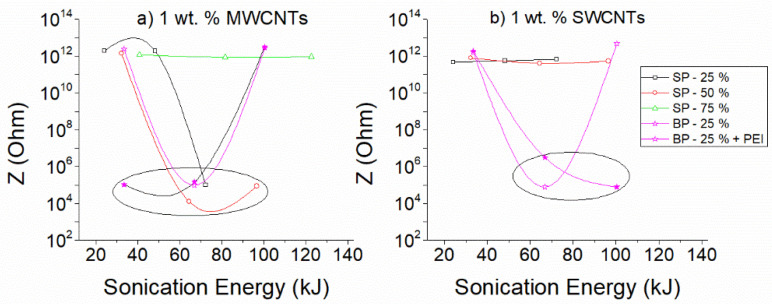
Variation of |Z| values at a frequency of 0.05 Hz as a function of sonication energy of aqueous dispersions of (**a**) MWCNTs 1 wt.% and (**b**) SWCNTs 1 wt.%. Note that SP stands for small and BP for big probe; P stands for PEI.

## Data Availability

Data available in request. The data presented in this study are available on request from the corresponding author.

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
