# Peer review of "The Use of Electrochemical Impedance Spectroscopy as a Tool for the In-Situ Monitoring and Characterization of Carbon Nanotube Aqueous Dispersions"

_nanomaterials, 2022, doi:10.3390/nano12244427_

Round 1
Reviewer 1 Report
The work presented is very interesting. Apart from individual typos, I think it can be published in its current version.
It raises a very interesting point of view of the application of EIS in synthesis optimization. Due to the difficulty of analyzing the EIS results, I believe that the authors have very responsibly fulfilled their task.
Author Response
Reviewer 1.
Reviewer’s Comment: The work presented is very interesting. Apart from individual typos, I think it can be published in its current version.
It raises a very interesting point of view of the application of EIS in synthesis optimization. Due to the difficulty of analyzing the EIS results, I believe that the authors have very responsibly fulfilled their task.
Author’s Response: The authors thank the reviewer for the response and the kind words.
Reviewer 2 Report
Authors have presented their work on utilizing EIS measurements to understand the effect of sonication on various CNT samples. As application of CNT depends on dispersion and proper alignment of nanotubes, this work has potential to guide an optimized way of dispersing CNT in solution for various practical application. In view of this, I recommend to publish this article after addressing my suggestions and questions:
1. How the author are defining conductivity as a function from impedance vs freq. plot. It should be presented in the form of ionic and electrical conductivity.
2. Imp. Vs Freq plot has various shapes. Authors should define the significance of the these specific shapes.
3. It is advised that nyquits plot should be included in the paper to support the impedance plot.
4. Authors are suggested to provide an example that how the optimized dispersion helps in final application of CNT.
Author Response
Reviewer’s Comment: 1. How the authors are defining conductivity as a function from impedance vs freq. plot. It should be presented in the form of ionic and electrical conductivity.
Author’s Response: The authors thank the reviewer for the comment. The analysis of the results in this work is based on the Complex Resistance, the Impedance Z. The real and imaginary part Z’(ω) & Z’’(ω) of the Impedance Z, can be employed to calculate the conductivity in the frequency domain via the following equation.
[see attached file]
where k is the geometrical constant of the material under testing.
The conductivity is the inverse of the Impedance measure. The analysis adopted by the authors in this paper is based on impedance as a complex quantity. The value of conductivity may be straightforwardly calculated, but to our point of view it will not add to the discussion of the paper to present the analysis in conductivity terms. For reasons of completeness, the aforementioned equation is included in the supporting information.
Reviewer’s Comment: 2. Imp. Vs Freq plot has various shapes. Authors should define the significance of the these specific shapes.
Author’s Response: The authors thank the reviewer for the comment. The significance of these shapes is defined in the abstract (please see line 20-24). ‘’ For dispersions above the percolation threshold a drop of |Z| approximately by seven orders of magnitude was observed followed by a linear reduction. The dramatic change in |Z| is regarded as an indication of the formation of a conductive path or a destruction of an already existing one during sonication and can be used to characterize the dispersion and morphology state of CNTs.’’ Above that, all the changes in the Impedance Z are discussed in section 3, Results and Discussion.
Reviewer’s Comment: 3. It is advised that nyquits plot should be included in the paper to support the impedance plot.
Author’s Response: The authors thank the reviewer for the comment. Nyquist plots were not included in the paper purely to avoid overloading the interested reader with data. So, the authors chose to present the EIS results through the Imp. Vs Freq. bode plots since in these plots all the information we wanted to highlight is clearly visible and further representation of the data would not significantly add to the discussion. However, for reasons of completeness we would be willing to include Nyquist plots preferably in the supplementary material section, if the reviewer considers them indispensable for the completeness of the presentation.
Reviewer’s Comment: 4. Authors are suggested to provide an example that how the optimized dispersion helps in final application of CNT.
Author’s Response: The authors thank the reviewer for the comment. Optimizing the dispersion of CNTs is expected to enhance the application of this type of nanomaterials in many ways making this type or material a sustainable and cost-efficient nano-reinforcement for Cementitious and polymeric matrices. This is already stressed in the manuscript since it is central to the scope of the presented work, and is also further analysed as following:
- Dispersion optimization is directly linked to performance optimization. It has been shown that an efficient dispersion of a nanophase within a matrix increases mechanical, thermal, and electrical properties. In addition, the creation of agglomerates is significantly reduced, thus decreasing the stress concertation at these sites which may lead to catastrophic failure (please see Introduction page 1, lines 40-43).
- An optimized and efficient dispersion significantly contributes to cost reduction. If the dispersion of a nanophase within a matrix is enhanced, the amount of reinforcing phase required to increase the mechanical performance or to achieve the electrical percolation threshold is significantly reduced. This directly is linked to costs involved with CNT utilization (NEW INSERTION, please see Introduction page 2, lines44-47)
- Sustainability: The more efficient a CNT dispersion the more sustainable the final material would be since the amount of the nanophase required to achieve a certain performance enhancement is significantly decreased. (NEW INSERTION, please see Introduction page 2, lines 48-50)

Reviewer 3 Report
A. Gkaravela and co-workers report an in situ monitoring and characterization for CNTs dispersion in aqueous media using EIS technology. The whole article is logical and has a strong scientific approach. In the EIS test section, the authors use different times to make comparisons and combine changes in morphology as a way to validate the hypothesis. While extensive tests in EIS have been done, there are still some problems should be solved. Before publication in Nanomaterials, this paper will need to be revised. Accordingly, my recommendation is Minor Revision. The following is my detailed review:
1. Why did the authors choose to test in the high frequency region rather than the mid and low frequency regions?
2. In Figure 5, the authors claim that higher ultrasound times and powers can damage the structure of carbon nanotubes. Could a finer threshold of time or frequency be provided? In addition, it is suggested that the authors add SEM tests to support that.
3. Electrochemical tests are very sensitive and small changes in the EIS test process can lead to some changes in the results. It is recommended that the authors add full test parameters including but not limited to Internal Resistance of the Solution, Amplitude, Quiet Time, and Sensitivity Scale Setting.
4. For a scientific article, the authors should pay attention to details in the manuscript, including the completeness of the author's signature, references (e.g. Ref 58), and issues of language presentation (e.g. Page 3 Line 47, Page 4 Line 177). It is recommended that authors consult a professional English editor for careful grammatical revision before publication.
Author Response
Reviewer’s Comment: 1. Why did the authors choose to test in the high frequency region rather than the mid and low frequency regions?
Author’s Response: The authors thank the reviewer for the comment. The impedance measurements were performed in the frequency region of 10-2 Hz to 105 Hz. This frequency window was chosen to cover and observe the most significant phenomena which take place during the interaction between material and alternating electric field. As should be noted, higher frequencies refer to more conductive media and are not relevant to this study. Additionally, this is the practical limit of our measuring platform, as mentioned in the experimental section (please see Materials and Methods, page 4, lines 178-180).
Reviewer’s Comment: 2. In Figure 5, the authors claim that higher ultrasound times and powers can damage the structure of carbon nanotubes. Could a finer threshold of time or frequency be provided? In addition, it is suggested that the authors add SEM tests to support that.
Author’s Response: The authors thank the reviewer for this insightful comment, which is directly relevant to the scope of this work. Using EIS, we may monitor both the destruction of the nanophase and its dispersion which prove to be antagonistic mechanisms, and this is what is being demonstrated in the paper. Figure 5 presents a re-agglomeration after extended sonication time at a specific amplitude which to our point of view is direct evidence of the phenomena that EIS shows. Further investigations using SEM could be performed in bucky papers to examine what is the effect in the microstructure (i.e., the aspect ratio of the CNTs), as is mentioned as possibility in the supporting information. However, we believe that the main conclusion can be detected very efficiently with EIS, is already verified optically, both in the macro (Figure S5) and micro (Figure 5) and no further verification is indispensable. In case the reviewer still believes that the SEM evaluation is needed we are willing to perform it.
Reviewer’s Comment: 3. Electrochemical tests are very sensitive and small changes in the EIS test process can lead to some changes in the results. It is recommended that the authors add full test parameters including but not limited to Internal Resistance of the Solution, Amplitude, Quiet Time, and Sensitivity Scale Setting.
Author’s Response: The authors thank the reviewer for the comment. Detailed parameters are added in the experimental section (please see Materials and Methods, page 4, lines 177-181).
Reviewer’s Comment: 4. For a scientific article, the authors should pay attention to details in the manuscript, including the completeness of the author's signature, references (e.g. Ref 58), and issues of language presentation (e.g. Page 3 Line 47, Page 4 Line 177). It is recommended that authors consult a professional English editor for careful grammatical revision before publication.
Author’s Response: The authors thank the reviewer for the comment. The problem that the reviewer rightly detected is due to the automatic reference formatting using an open access software. All references are checked and corrected in the revised manuscript. The manuscript is also carefully edited for the English Language.
Reviewer 4 Report
The article is written on a topical topic - the development of methods for controlling the processes of creating (filler dispersion) of filled composites. Such materials have unique properties, but the quality of their production is difficult to control. The proposed method is original.
1. The first section provides a fairly detailed literature review on the creation of new smart materials based on carbon nanotubes and the related problem of providing the necessary filler dispersion. As a method for controlling the process of ultrasonic dispersion, the authors propose the EIS method.
The review is quite detailed and based mainly on modern literature. A really urgent problem has been formulated. The selected monitoring technique is original.
2. The second section presents the materials and research methods used. They are chosen adequately, are modern and correspond to the purpose of the work.
3. The third section presents the main results obtained by the authors. The illustrations are well done. The description of the data is quite detailed. When describing the results, a short discussion of them is also given.
4. The fourth section contains conclusions. This section needs significant revision. About this in the comments.
5. A large file with accompanying data has also been added to the text of the article, which improves the overall impression of the work.
Notes:
1. The fourth section is written as a very good discussion of the results! Therefore, I recommend that it be titled "discussion" and additionally write briefly a few paragraphs with the main conclusions on the results obtained.
Author Response
Reviewer’s Comment: 1. The fourth section is written as a very good discussion of the results! Therefore, I recommend that it be titled "discussion" and additionally write briefly a few paragraphs with the main conclusions on the results obtained.
Author’s Response: The authors thank the reviewer for the comment. The conclusion section was rewritten in a more concise way and remaining information was duly moved to the discussion section.
Round 2
Reviewer 4 Report
The authors took into account my remark and made the necessary corrections.
I believe that in its current form the article can be accepted for publication.